Possible eucynodont (Synapsida: Cynodontia) tracks from a lacustrine facies in the Lower Jurassic Moenave Formation of southwestern Utah

Hurtado Holly 1 2
Harris Jerald D. jerry.harris@utahtech.edu 1
Milner Andrew R.C. 2
1 Earth and Environmental Sciences, Utah Tech University , St. George , UT , United States of America
2 St. George Dinosaur Discovery Site , St. George , UT , United States of America
Marsicano Claudia
Electronic publication date: 2024 Jun 26
Publication date: 2024
Volume: 12
Electronic Location ID: e17591
Received 2023 Jul 24; Accepted 2024 May 28
Copyright: ©2024 Hurtado et al.
Copyright year: 2024
Copyright holder: Hurtado et al.
License: This is an open access article distributed under the terms of the Creative Commons Attribution License, which permits unrestricted use, distribution, reproduction and adaptation in any medium and for any purpose provided that it is properly attributed. For attribution, the original author(s), title, publication source (PeerJ) and either DOI or URL of the article must be cited.
License URL: https://creativecommons.org/licenses/by/4.0/

Keywords: Ichnology, Footprint, Utah, Jurassic, Moenave, Cynodont, Synapsid

Funding: The authors received no funding for this work.

==============================
Eight fossil tetrapod footprints from lake-shore deposits in the Lower Jurassic Moenave Formation at the St. George Dinosaur Discovery Site (SGDS) in southwestern Utah cannot be assigned to the prevalent dinosaurian (Anomoepus, Eubrontes, Gigandipus, Grallator, Kayentapus) or crocodyliform (Batrachopus) ichnotaxa at the site. The tridactyl and tetradactyl footprints are incomplete, consisting of digit- and digit-tip-only imprints. Seven of the eight are likely pes prints; the remaining specimen is a possible manus print. The pes prints have digit imprint morphologies and similar anterior projections and divarication angles to those of Brasilichnium, an ichnotaxon found primarily in eolian paleoenvironments attributed to eucynodont synapsids. Although their incompleteness prevents clear referral to Brasilichnium, the SGDS tracks nevertheless suggest a eucynodont track maker and thus represent a rare, Early Mesozoic occurrence of such tracks outside of an eolian paleoenvironment.

Introduction

Fossil tetrapod tracks attributed to pre-Cenozoic synapsids have been found throughout the southwestern United States in strata ranging from Lower Permian (e.g., McKeever & Haubold, 1996; q.v., Marchetti et al., 2019) through Upper Cretaceous (Lockley & Foster, 2003). The vast majority of these tracks, particularly from the Early Permian and Late Triassic–Early Jurassic, occur in eolian facies (the Chelichnus ichnofacies of Hunt & Lucas, 2006a; Hunt & Lucas, 2006b; q.v., Krapovickas et al., 2016). In Utah specifically, such tracks are common in the eolian, Upper Triassic–Lower Jurassic Nugget Sandstone and correlative (per Sprinkel, Kowallis & Jensen, 2011) Wingate Sandstone and Navajo Sandstone (Lockley, 2011; Lockley & Hunt, 1995; Lockley et al., 2004; Lockley et al., 2011; Tweet & Santucci, 2015; Engelmann & Chure, 2017); they have also been reported from the roughly correlative Aztec Sandstone of California and Nevada (Reynolds, 2006; Rowland & Mercadante, 2014). In contemporaneous, non-eolian strata in the southwestern United States, non-synapsid tetrapod tracks otherwise predominate, while synapsid (specifically, at this time, dicynodonts, non-mammaliaform eucynodonts (sensu Hopson & Kitching, 2001), and mammaliaforms (sensu Rowe, 1988; q.v. Sereno, 2006)) tracks are rare (e.g., Hunt & Lucas, 2006a; Hunt & Lucas, 2006b; Klein & Lucas, 2021; Lockley & Gierliński, 2006; Lockley & Gierliński, 2014; Lockley, Kirkland & Milner, 2004). However, non-synapsid tetrapod tracks also are relatively common in eolian settings in the region (e.g., Baird, 1980; Bennett, Harris & Milner, 2023; Hamblin, Bilbey & Hall, 2000; Lockley et al., 2014; Lockley et al., 2021a; Lockley et al., 2021b; Milàn, Loope & Bromley, 2008; Milner et al., 2011; Milner et al., 2023a; Stokes & Madsen Jr, 1979). Furthermore, burrows attributed to synapsids are also known exclusively from eolian facies in this region (e.g., Hasiotis, Parrish & Chan, 2019; Odier, 2006; Riese, Hasiotis & Odier, 2011). Taken at face value, this regional disparity in the distribution of synapsid ichnofossils with respect to non-synapsid tetrapod tracks leads to the conclusion that Early Mesozoic synapsids in the American Southwest preferentially inhabited eolian environments; these data do not suggest a preservational bias for synapsid ichnites over those of non-synapsid tetrapods. Yet synapsid body fossils are known from Upper Triassic and Lower Jurassic, non-eolian strata in the same region (e.g., Kligman et al., 2020; Jenkins Jr, Crompton & Downs, 1983; Sues & Jenkins Jr, 2006), indicating that the eolian ichnological record cannot be the complete story, and that synapsid tracks should occur in other environments as well. Why they are thus rarer in contemporaneous non-eolian facies is unknown.

The St. George Dinosaur Discovery Site (SGDS) in Washington County, southwest Utah (Fig. 1) preserves an abundant and moderately diverse ichnofauna in lacustrine and marginal lacustrine environments (the Grallator ichnofacies of Hunt & Lucas, 2006a; Hunt & Lucas, 2006b) of the Whitmore Point Member of the Moenave Formation, including invertebrate, fish, and sauropsid tracks and trails (Milner et al., 2011). Possible synapsid tracks at the site have been briefly mentioned (Milner, Lockley & Johnson, 2006; Milner et al., 2011), but until now have not been studied in detail. Their tentative synapsid attribution stemmed from a combination of their small size; a similarity to some tracks referred to the ichnotaxon Brasilichnium, which has long been attributed to synapsids; and a general inability to refer them to any of the sauropsid ichnotaxa at the site, in the region, and from the earliest Jurassic.

Figure 1 Location (white stars) of the St. George Dinosaur Discovery Site (SGDS) in Washington County, St. George, Utah.

(A) General location in southwestern Utah. (B) Location superimposed on a geologic map of the vicinity. For more details about the geology of the locality, see Kirkland & Milner (2006).

Geological setting

Most of the fossils at the SGDS are from the lowermost Jurassic (Suarez et al., 2017) Whitmore Point Member of the Moenave Formation. This unit overlies the Dinosaur Canyon Member, which contains the Triassic–Jurassic boundary, and is overlain by the Springdale Sandstone Member at the base of the Kayenta Formation (Kirkland & Milner, 2006; Kirkland et al., 2014; Fig. 2). The Whitmore Point Member consists of multiple fossiliferous horizons, the most prominent and fossiliferous of which is the Johnson Farm Sandstone Bed (Kirkland et al., 2014). The Johnson Farm Sandstone Bed is itself divided into a lower Johnson Farm Main Track Layer, a lower–middle Johnson Farm Split Track Layer, and several thinly bedded, apparently conformable, fine-grained-sandstone Top Surface horizons (Fig. 2), all of which preserve abundant vertebrate tracks, invertebrate traces, sedimentary structures, and rare body-fossil remains (Milner, Lockley & Johnson, 2006; Milner et al., 2011). The fossils and sedimentary structures reveal the paleoenvironment of the Johnson Farm Sandstone Bed as having been deposited along the shore of Lake Whitmore (formerly Lake Dixie), a large freshwater lake (Kirkland & Milner, 2006; Kirkland et al., 2014; Milner, Lockley & Johnson, 2006; Tanner & Lucas, 2009). The stratigraphy and sedimentology of the SGDS site, as well as the Whitmore Point Member of the Moenave Formation across the region, have been detailed previously (Kirkland & Milner, 2006; Kirkland et al., 2014).

Figure 2 Stratigraphic section at and immediately around the St. George Dinosaur Discovery Site in St. George, Utah.

Possible synapsid tracks SGDS 18 and 190 come from the Top Surface Tracksite horizon of the Johnson Farm Sandstone Bed (red arrows). For more details on the stratigraphy of the site, see Kirkland & Milner (2006); for more details on the stratigraphy of the site in the broader context of the Whitmore Point Member of the Moenave Formation across the region, see Kirkland et al. (2014).

Among tracks preserved on the Top Surface horizons at the SGDS specifically, the vast majority were made subaerially rather than subaqueously. The horizons generally preserve 2D and/or 3D ripple marks, indicating they were subaqueous at various times, but the morphologies of most of the common Batrachopus, Eubrontes, and Grallator tracks superposed on the ripples across the surface are typical of subaerially registered tracks. A depression on the topographically irregular Top Surface preserved at the SGDS, located well away from the region in which the in situ tracks described herein lie, either contained subaerial, saturated sediment or was under shallow water based on the “sloppy” morphologies of Grallator tracks preserved therein, and a few small Batrachopus trackways on the Top Surface at the SGDS include transitions from swim to walk and walk to swim as they entered or exited shallow-water-filled swales in the Top Surface (Milner, Lockley & Kirkland, 2006), but the majority of the surface was otherwise subaerial at the time of track formation. The surface was, however, periodically affected by wave action from Lake Whitmore. A more detailed interpretation of the paleoenvironment of the Top Surface at the SGDS was presented by Kirkland & Milner (2006).

The tracks described herein all come from the Top Surface horizons and are in situ at the SGDS, except SGDS 190, which is ex situ and on display at the SGDS.

Materials and Methods

Measurements of the tracks (Fig. 3) described here were taken using digital calipers on the specimens or replicas of the specimens. The divarication angles between digit imprints were measured using photographs taken orthogonal to the planes in which the specimens lay, using a protractor between straight lines drawn through the long axes of the imprints. For curved digit imprints, the long axes used in divarication measurements were straight lines drawn through the proximal, not distal, ends of the imprints. Photos were taken with a Nikon D5200 digital camera outfitted with an AF-S Nikkor 18–140 mm VR lens under artificial lighting. Photogrammetric images used to generate the digital elevation model (DEM) were taken with a Nikon D870 digital camera outfitted with an AF-S Micro Nikkor 40 mm lens. General photogrammetric methods used follow Bennett (2022); the DEM was generated in Agisoft Metashape v. 2.0.3.

Figure 3 Schematic depicting how measurements of possible eucynodont tracks SGDS 18 and 190 were taken.

Diagrams use a tracing of SGDS 18-T7 as a model. (A) Track measurements: dl, digit length; tl, track length; tw, track width. (B) Measurements of divarication angles between individual digit traces (II–V).

Specimen SGDS 190 was scanned November 13–14, 2023 using a Faro Model 14000 Edge laser scanning arm with a build accuracy of 0.023 mm, calibrated on November 10, 2023 with a spatial error of <0.006 mm. The software used to digitally acquire scan data from the 14000 Edge was PolyWorks 2018 Metrology Suite IR3.1 64-bit. Scan data were further processed in the PolyWorks IMAlign workspace to remove background objects and combine different scans of the specimen to ensure complete coverage. The scan data were compiled into a polygonal model using the PolyWorks IMMerge tool; the resulting model then was “cleaned” (spikes and non-model data removed) in the PolyWorks Modeler tool. The resulting .obj file was further processed in Geomagic Wrap 2017.0.0 64-bit for final cleaning. The model was converted to a point model to eliminate point data discrepancies, then digitally wrapped to create a single manifold model. The hole-fill tool was used to fill any holes in the resulting polygonal model, and then the MeshDoctor tool was used to confirm that the model was a single manifold object without microscopic or invisible errors. This was then exported as a model in .obj format for end use.

Herein we follow Minter, Braddy & Davis (2007) by using the term “imprint” when discussing a discrete, non-continuous trace, such as a digit or sole imprint, and the term “impression” when discussing a more continuous trace. Although this descriptive system was developed for use with arthropod, rather than vertebrate, traces, the system is useful for describing vertebrate traces as well; its adoption here is simply for the sake of clarity.

All specimens described herein are reposited at the SGDS in St. George, Utah. A replica of SGDS 190 is reposited in the University of Colorado Museum in Boulder, Colorado as specimen UCM 177.37; all other specimen replicas are retained at the SGDS.

Descriptions

As far as is currently known, all Early Mesozoic synapsids had pentadactyl manus and pedes. None of the tracks described herein, however, possess five digit imprints, making determining which imprints correspond to which digits impossible. Herein we number the digit imprints using the system common to other tetrapods in which digit lengths increase from digits I–IV and decrease again in digit V (i.e., ectaxony). We acknowledge, however, that these relative digit proportions may not apply to any or all Early Mesozoic synapsids (see below), and that proportions of digit imprints made by at least some of these taxa are complicated by their apparent possession of digital arcades (Kümmell & Frey, 2012).

All the tracks described herein are natural molds (concave/negative epirelief).

SGDS 190 (Fig. 4, Supplemental Information 1)

SGDS 190 is a single, ectaxonic right track that comprises four moderately deep digit imprints only, here interpreted as digits II–V using the aforementioned reasoning (Table 1). The imprints lack discernible digital pad and claw traces. Digit imprints II and III are more diamond-shaped, but still rounded distally. Imprints IV and V do not taper either proximally or distally; instead, they have rounded proximal and distal ends that are approximately the same widths as the midpoints of the imprints. The imprints of digits II and III are parallel and straight; the proximal end of the imprint of digit IV parallels those of II and III, but distally the imprint curves laterally to parallel the short and straight imprint of digit V. The curvature of digit IV could be due to some slipping in wet sediment when the track maker pushed off. SGDS 190 lacks a clear sole mark, but the outermost margins of the outermost digit imprints angle inward toward the bases of the other digit imprints in such a way as to suggest the sole region was short anteroposteriorly.

Figure 4 Possible eucynodont track SGDS 190 from the Lower Jurassic Moenave Formation of St. George, Utah.

Scale bar in cm.

Table 1 Measurements for possible eucynodont tracks from the Lower Jurassic Moenave Formation of St. George, Utah.

Track	Total Length (mm)	Total Width (mm)	Digit II length (mm)	Digit III length (mm)	Digit IV length (mm)	Digit V length (mm)	∠ II–III (°)	∠III–IV (°)	∠IV–V (°)	∠Outer-most digits (°)	
190	13.0	15.1	8.6	9.7	10.5	5.8	4.0	27.0	29.0	59.0	
18-T3-1	18.7	19.9	n/a	5.1	15.5	9.6	n/a	20.0	12.0	32.0	
18-T3-2	12.8	33.9	7.7	11.7	8.2	6.5	?	36.5	24.0	?	
18-T3-3	16.0	26.5	5.6	7.8	11.0	7.2	11.0	14.0	37.0	61.0	
18-T3-3-2	11.8	15.6	n/a	8.0	9.0	n/a	n/a	32.0	n/a	n/a	
18-T3-4	14.5	27.0	2.4	5.7	8.4	9.5	4.0	18.0	21.0	43.0	
18-T3-4-2	12.0	27.5	n/a	6.0	6.9	6.0	7.0	11.0	9.0	28.0	
18-T3-5	23.1	23.0	n/a	9.0	7.3	7.9	n/a	32.0	22.0	53.0	
18-T3-5-2	18.7	16.9	n/a	7.2	9.3	6.3	n/a	13.0	17.0	29.0	
18-T3-6	12.8	24.6	n/a	12.3	10.4	6.7	n/a	53.0	31.0*	24.0	
18-T7	10.0	21.7	7.5	8.5	7.8	6.0	8.0	67.0	24.0	98.0	
Notes.

∠ divarication angle

* angle anterior, rather than posterior, to track

? one digit impression too vague to accurately determine axis

n/a not applicable

SGDS 18-T3 (Figs. 5, 6, 7 and 8)

SGDS 18-T3 is a short trackway of five apparent pes prints (SGDS 18-T3-1, -2, -3, -4, & -6) and one possible manus print (SGDS 18-T3-5) (Fig. 5). Tracks T3-1, -2, and -6 have a slight outward rotation from the trackway axis.

Figure 5 (A) Digital elevation model, and (B) schematic depicting relative positions of possible eucynodont tracks 1–6 in trackway SGDS trackway 18-T3.

Scale bar = 5 cm. See Figs. 6–8 for individual track details.

Figure 6 Possible eucynodont tracks SGDS 18-T3-1 and 18-T3-2 from the Lower Jurassic Moenave Formation of St. George, Utah.

(A) SGDS 18-T3-1. (B) SGDS 18-T3-2. Scale bars = 1 cm.

Figure 7 Possible eucynodont tracks SGDS 18-T3-3, -4, and -5 from the Lower Jurassic Moenave Formation of St. George, Utah.

Scale bar in cm.

Figure 8 Possible eucynodont track SGDS 18-T3-6 from the Lower Jurassic Moenave Formation of St. George, Utah.

Scale bar in cm.

SGDS 18-T3-1 (Fig. 6A), an apparent mesaxonic right pes print, has three subparallel digit imprints, herein interpreted as digits III–V, The imprints are straight, narrow, and roughly oblong, tapering distally; digit IV, the longest and centrally placed imprint, tapers proximally as well. The distal tapers of the imprints suggest short claws. The proximal ends of the imprints all lie approximately at the same level. As with SGDS 190, the digit imprints lack discernible digital pads.

SGDS 18-T3-2 (Fig. 6B), an apparent mesaxonic left pes print, has four imprints, herein interpreted as digits II–V. Unlike those of 18-T3-1, the imprints divaricate markedly (Table 1). The imprint of digit V is short and curved outward; the other imprints are straight. The imprint of digit II is faint and short. The both proximally and distally tapering imprint of digit III is the deepest and most pronounced trace; it is also the longest and most distally extended, unlike the apparent pattern in 18-T3-1. The imprints of digits IV and V have rounded proximal and distal ends. As with 18-T3-1, all imprints lack discernible digital pads. Both 18-T3-1 and 18-T3-2 possess longer digit imprints than the rest of the tracks in the trackway, but otherwise are similar in relative digit lengths and by tapering on the distal ends.

SGDS 18-T3-3, 18-T3-4, and 18-T3-5 (Fig. 7) are unusual, comprising primary tracks connected to secondary sets of imprints by clear, linear, but shallow, drag impressions (Figs. 5 and 7), each spanning roughly 30 mm between their primary and secondary tracks. The primary track of SGDS 18-T3-3, a paraxonic to ectaxonic right pes print, comprises four short, relatively narrow digit-tip imprints (II–V) that, as in SGDS 190, increase in length from digit II to IV and decrease again in digit V, but unlike in SGDS 190, the imprint of digit II is the shortest (Table 1). Digit imprints II and III are subparallel, but angle inward whereas digit imprints IV and V angle outward, giving the track a paraxonic sense. The imprint of digit IV tapers distally into what may be a short claw trace, but the remaining digit imprints are rounded distally. The secondary track of SGDS 18-T3-3 (labeled 18-T3-3-2 in Table 1) comprises imprints only of digits III and IV; unlike their primary-track counterparts, these imprints taper distally. Their angulations mirror those of their primary-track counterparts.

The primary track of left pes SGDS 18-T3-4 resembles 18-T3-3 except that its imprints are shorter and all rounded distally, though that of digit IV is still the longest of the set. The secondary track of 18-T3-4 (labeled 18-T3-4-2 in Table 1) comprises imprints of all four digits, but they are shorter and shallower than those of 18-T3-3. Unlike in 18-T3-3, the secondary imprints of 18-T3-4 appear rounded distally. Tracks 18-T3-3 and 18-T3-4 lie close to their trackway midline.

The primary track of SGDS 18-T3-5 lies lateral and slightly anterior to 18-T3-4, a position that suggests it might be a manus print. However, its morphology differs markedly from those of 18-T3-3 and 18-T3-4. The primary track comprises three faint, narrow, and shallow digit-tip imprints, likely those of digits III–V. Unlike those of 18-T3-3 and 18-T3-4, each roughly triangular imprint tapers sharply to a point distally, suggesting they may be claw traces. Rather than lying in an approximate, shallowly arcuate row, as do the imprints of 18-T3-3 and 18-3-4, the imprints of 18-T3-5 lie at markedly different levels with respect to each other, with the imprint of digit IV lying far anterior to the imprints of digits III and V. Their configuration is reminiscent of a Grallator theropod dinosaur track, but the close association of 18-T3-5 with 18-T3-4, plus the drag impressions and secondary print shared with 18-T3-3 and 18-T3-4, strongly suggest it was made by the same track maker as the other SGDS 18-T3 tracks. The imprints of digit III and V curve slightly distally in opposing directions. The secondary track of 18-T3-5 (labeled 18-T3-5-2 in Table 1) is virtually identical to the primary track, but fainter.

SGDS 18-T3-6 (Fig. 8) is a single apparently right pes print that somewhat resembles SGDS 190 and 18-T3-1; it is virtually the same size as SGDS 190. It comprises three distally tapering digit imprints, presumably of digits III–V. The imprints of digit III and V are straight; that of digit IV curves slightly outward at its tip. Swollen, rounded areas immediately proximal to the tapered claw imprints could be digital pad imprints.

SGDS 18-T7 (Fig. 9)

SGDS 18-T7 is a single, tetradactyl, apparent paraxonic to ectaxonic right pes print that is similar to SGDS 190, 18-T3-3, and 18-T3-4 in overall morphology. As in SGDS 18-T3-3 and T3-4, the subequal imprints of digits II and III are subparallel to each other and angle inward; the shorter imprints of digits IV and V are subparallel to each other and angle outward, giving the track a paraxonic sense. Digit imprints IV and V appear to curve outward slightly at their distal ends. All four imprints taper toward their distal ends, which suggests the presence of claws. No obvious digital pad imprints are discernible.

Figure 9 Possible eucynodont track SGDS 18-T7 from the Lower Jurassic Moenave Formation of St. George, Utah.

Scale bar in cm.

Comparisons

The mostly tetradactyl SGDS tracks described herein are markedly unlike the dominantly tridactyl, mesaxonic ornithischian (Anomoepus) and theropod (Eubrontes, Gigandipus, Grallator, Kayentapus) dinosaur tracks, all made primarily by bipedal track makers, known from Late Triassic–Early Jurassic of North America and that are abundant at the SGDS and in its geographic and stratigraphic vicinity. Anomoepus can include manus prints, demonstrating facultative quadrupedality by their track makers, but manus tracks are pentadactyl and entaxonic (Olsen & Rainforth, 2003), and therefore unlike the SGDS tracks described herein. Additionally, Anomoepus manus prints unassociated with pes prints are unknown and unexpected given that the weight-bearing hind limbs of presumed Anomoepus track makers (facultatively quadrupedal basal ornithischian dinosaurs) would be expected to register much deeper and more pronounced tracks than the manus. Late Triassic–Early Jurassic sauropodomorph ichnotaxa (Eosauropus, Evazoum, Kalosauropus, Otozoum, Navahopus, Pseudotetrasauropus), while being tetradactyl to pentadactyl, are all far larger than the SGDS tracks described herein; they also have markedly different digit and sole imprint morphologies and proportions (Lallensack et al., 2017; Mukaddam et al., 2020; Rainforth, 2003), and thus can be readily excluded as possible referrals for the SGDS tracks.

The number of Mesozoic, non-dinosaurian tetrapod ichnotaxa to which the SGDS tracks described herein could be compared is substantial. We limit our comparisons to ichnotaxa known from Upper Triassic–Early Jurassic strata of the western United States because those are temporally and geographically the closest to the SGDS tracks and therefore the most likely to be possibly congeneric. We do not include comparisons to various Paleozoic (e.g., temnospondyl, such as Batrachichnus) and Triassic (e.g., chirotheriid, Gwyneddichnium) ichnotaxa that are currently understood to not extend into the Jurassic (e.g., Klein & Lucas, 2021), or to Triassic taxa (e.g., kuehneosaurids, tanystropheids) that likewise are not currently understood to extend into the Jurassic, though we include comparisons to Triassic purported synapsid tracks.

BatrachopusHitchcock, 1845 (Fig. 10A)

Batrachopus is particularly important to compare to the SGDS tracks described herein because it is one of the most common tetrapod ichnotaxa at the SGDS. The ichnotaxon is attributed to early crocodyliforms, such as Protosuchus (Olsen & Padian, 1986). Batrachopus tracks are found in Lower Jurassic strata of France (Moreau et al., 2019), the northeastern (Hitchcock, 1845; Olsen & Padian, 1986) and southwestern (Lockley, Kirkland & Milner, 2004; Lockley et al., 2018) United States, and possibly southern Africa (Lockley, Kirkland & Milner, 2004; Lockley et al., 2018) and Colombia (Mojica & Macia, 1987), as well as Middle?–Upper Jurassic strata of Morocco (Masrour et al., 2020) and Lower Cretaceous strata of South Korea (Kim et al., 2020). Batrachopus manus tracks are pentadactyl (though often tridactyl or tetradactyl, as well) with varying digit orientations: usually the digit imprints are spread such that digit II points anteriorly, digit IV points laterally, and digit V points posteriorly (Olsen & Padian, 1986), but numerous referred specimens have more variable digit orientations, including having digit imprints with low divarication angles. Digit imprints are typically short but wide and may or may not terminate in narrower claw imprints. Batrachopus pes tracks are ectaxonic and tetradactyl (digits I–IV), with digit III being the longest. Digit V, if present, consists of an oval imprint behind that of digit III. Digit imprints are longer than those of the manus, but also relatively wide. Batrachopus trackways demonstrate that the manus and pes prints rotate markedly outward. Numerous Batrachopus tracks at SGDS fit this general description and differ markedly from the tracks described in this paper. However, we note that the sheer diversity of track morphologies that have been attributed to Batrachopus renders comparisons to this ichnotaxon somewhat problematic, and strongly suggests that it requires detailed and updated review and revision.

Figure 10 Schematic morphological comparisons between manus and pes prints of (A) Batrachopus, (B–H) Mesozoic synapsid ichnotaxa and (I–N) various non-synapsid ichnotaxa and tracks of extant tetrapod taxa (not to scale).

(A) Composite Batrachopus from the Lower Jurassic Moenave Formation, Arizona (traced from Olsen & Padian, 1986). (B) Ameghinichnus from the Middle Jurassic La Matilde Formation, Santa Cruz, Argentina (traced from De Valais, 2009). (C) Navahopus from the Lower Jurassic Navajo Sandstone, Arizona, USA (traced from Baird, 1980). (D) Pentasauropus from the Middle Triassic Cerro de las Cabras Formation, Mendoza, Argentina (traced from Lagnaoui et al., 2019). (E) Therapsipus from the Middle Triassic Holbrook Member of the Moenkopi Formation, Arizona, USA (traced from Hunt et al., 1993). (F) Dicynodontipus (“Gallegosichnus” type) from the Upper Triassic Vera Formation, La Rioja, Argentina (traced from Melchor & De Valais, 2006). (G) Dicynodontipus (“Calibarichnus” type) from the Upper Triassic Vera Formation, La Rioja, Argentina (traced from Melchor & De Valais, 2006). (H) Brasilichnium from the Lower Cretaceous Botucatu Formation, São Paulo, Brazil (traced from Fernandes & De Souza Carvalho, 2008). (I) Exocampe from the Portland Formation (Newark Supergroup), Massachusetts, USA (traced from Lull, 1915). (J) “Conventional” modern lizard (Iguana) tracks (traced from Diedrich, 2005). (K) Emydhipus turtle tracks from the Upper Jurassic Lastres Formation, Asturias, Spain (traced from Avanzini et al., 2005). (L) Modern salamander (western newt, Taricha) tracks made in wet mud (traced from Brand, 1996). (M) Modern salamander (western newt, Taricha) tracks made in subaqueous sand at a 25° angle (traced from Brand, 1996). (N) Modern bullfrog (Lithobates) tracks (traced from Tkaczyk, 2015).

The relative narrowness and separation of digit imprints, as well as the low divarication angles between digit imprints, of all of the SGDS tracks described herein, except for 18-T3-2, preclude them from being classic Batrachopus manus prints. Additionally, the isolated SGDS tracks (190, 18-T7) are unlikely to be manus prints because inferred Batrachopus track makers (protosuchian crocodyliforms) would likely have left more pronounced pes than manus prints, making the absence of associated pes prints with these SGDS tracks bizarre. Also except for 18-T3-2, the anterior projections of the digit imprints of the tetradactyl SGDS tracks (190, T3-3, T3-4, 18-T7) are proportionately more subequal than those of Batrachopus pes tracks, in which the innermost imprint (digit I) is much shorter than the other digit imprints (Lockley et al., 2018; Moreau et al., 2019; Olsen & Padian, 1986). The tridactyl SGDS tracks (T3-1, T3-6) are more equivocal in this regard, although T3-1 and T3-6 are part of the T3 trackway, and thus associated with tetradactyl tracks T3-3 and T3-4, so they can be inferred to have had similar overall digit imprint proportions had they been tetradactyl. The absence of distinct claw imprints in the relatively deeply impressed SGDS 190, plus its inferred short sole imprint, also further distinguish that specimen from Batrachopus pes tracks. Overall, a case for attributing the SGDS tracks to Batrachopus is not well supported.

AmeghinichnusCasamiquela, 1961 (Fig. 10B)

Ameghinichnus tracks are usually attributed to mouse-sized mammaliaforms and have been found in Upper Triassic–Middle Jurassic strata of Argentina (Casamiquela, 1961; De Valais, 2009), South Africa (Olsen & Galton, 1984), Poland (Gierliński, Pieńkowski & Niedźwiedzki, 2004), and the western (Lockley et al., 2004) and possibly eastern (Olsen, 1988; Olsen & Rainforth, 2001) United States. Classic Ameghinichnus tracks, as described by De Valais (2009), comprise quadrupedal trackways with pentadactyl manus and pes tracks that are wider than long. Digit imprints II–V are subequal in length; all digit imprints lack claw imprints (except in possible specimens from the Newark Supergroup; Olsen & Rainforth, 2001, fig 59A) and are rounded and swollen distally, making them wider than more proximal parts of their imprints. Symmetrical manus tracks have subequal divarication angles between digit imprints; pes tracks have markedly greater divarication angles between digits I–II and IV–V than between II–III and III–IV. Thus, in both the manus and pes tracks, the digit imprints splay markedly, and are not subparallel. In A. patagonicus, smaller manus tracks lie medial to the pes tracks and are rotated inward, toward the midline, while the somewhat larger pes prints are rotated outward, away from the midline. Both manus and pes tracks are wider than they are long. Most Ameghinichnus tracks have distinct sole imprints.

The SGDS tracks described herein are all tridactyl or tetradactyl and lack sole imprints, unlike Ameghinichnus. Some of the SGDS tracks described herein possess distally tapering digit imprints, also unlike Ameghinichnus; those that lack claw imprints and are rounded distally lack distal swellings (though the digit IV imprint of SGDS 190 comes close), also unlike classic Ameghinichnus. The SGDS tracks generally lack the consistent splay (divarication angles) exhibited by Ameghinichnus tracks, sometimes possessing subparallel digit imprints. Tracks in trackway SGDS 18-T3 do not display the degrees of rotation that tracks in Ameghinichnus trackways do, and the possible manus track in this trackway lies lateral to the pes track, opposite the configuration in Ameghinichnus. Thus, the SGDS tracks do not fit within the Ameghinichnus paradigm.

NavahopusBaird, 1980 (Fig. 10C)

Navahopus is an uncommon ichnotaxon thus far reported exclusively from Lower Jurassic strata of the southwestern United States (Baird, 1980; Hunt & Lucas, 2006c; Milàn, Loope & Bromley, 2008; Reynolds, 2006). The Navahopus track maker is unclear: the tracks have been attributed to sauropodomorph dinosaurs (Baird, 1980; Milàn, Loope & Bromley, 2008) and large therapsid synapsids (Hunt & Lucas, 2006c; Lockley & Hunt, 1995; Shibata, Matsukawa & Lockley, 2006). Navahopus manus tracks are tridactyl, with two short, anteriorly oriented digit imprints and a large, laterally oriented, “falciform” claw imprint (Baird, 1980; Milàn, Loope & Bromley, 2008; q.v., Hunt & Lucas, 2006c). The manus imprints are mediolaterally elongate but anteroposteriorly short. Navahopus pes tracks are functionally tetradactyl with all digits rotated slightly laterally; all digit imprints taper distally, terminating in claw marks. They possess pronounced, posteriorly convex, though irregularly shaped sole imprints.

In addition to its much greater size, Navahopus morphology is distinctly different from those of the SGDS tracks. None of the SGDS tracks resemble Navahopus manus tracks, possessing more digits and lacking the “falciform” pollex claw imprint. The SGDS tracks lack the distinct sole imprint of Navahopus pes prints, but are similar in generally possessing subparallel digit imprints. However, the digit imprints of Navahopus are quite thin and distally tapering, while the SGDS track digit imprints are mostly wider, even if they taper distally. The SGDS tracks thus cannot be readily referred to Navahopus.

PentasauropusEllenberger, 1970 (Fig. 10D)

Pentasauropus manus and pes tracks are, as their name implies, pentadactyl, and similar in size and morphology (D’Orazi Porchetti & Nicosia, 2007). The originally described specimens from Upper Triassic strata of Lesotho (Ellenberger, 1970; Ellenberger, 1972), as well as specimens from Lower Triassic strata of Argentina (Citton et al., 2018) and Upper Triassic strata of western North America (Gaston et al., 2003; Lockley & Hunt, 1995) and Argentina (Marsicano & Barredo, 2004), consist almost exclusively of small, generally ovoid digit-tip imprints arranged roughly equally spaced in an anteriorly convex, arcuate pattern; the digit-tip imprints are generally, but not universally, wider than long. Other Late Triassic specimens from Argentina, as well as subsequently discovered specimens from the Middle Triassic of Argentina (Lagnaoui et al., 2019), also include large, mediolaterally wide, oval-, kidney-, or D-shaped palm/sole imprints that are loosely connected or entirely unconnected to the digit imprints. Pentasauropus trackways are wide gauge and have low pace angulation values. Although initially referred by Ellenberger (1970) to amphibians and sauropodomorph dinosaurs, and by Haubold (1984) to a sauropod or therapsid, Pentasauropus has more typically been attributed to dicynodont therapsids (D’Orazi Porchetti & Nicosia, 2007; Kammerer, 2018; Olsen & Galton, 1984), an interpretation supported by their restriction to Triassic strata.

Although size is a poor ichnotaxobase, known Pentasauropus tracks dwarf the SGDS specimens. While some of the SGDS tracks described herein similarly consist of digit-tip-only imprints, the imprints are all longer than wide and not generally distributed in the neat arc seen in Pentasauropus; the other SGDS tracks described herein consist of more elongate and narrow digit imprints, thus also differing from Pentasauropus. Tracks in the SGDS 18-T3 trackway have higher pace angulation values than do Pentasauropus trackways. The SGDS tracks do not pertain to Pentasauropus.

TherapsipusHunt et al., 1993 (Fig. 10E)

Therapsipus tracks, thus far described only from the Middle Triassic of Arizona, were made by a large, wide-bodied quadruped (Hunt et al., 1993). Tracks are tetradactyl to pentadactyl and consist of short, wide, typically blunt digit imprints connected to anteroposteriorly short but mediolaterally wide palm/sole imprints; this connection, plus the morphologies of the digits and palm/sole imprints and a greater degree of heteropody between the manus and pes, differentiate this ichnotaxon from Pentasauropus. Nevertheless, like Pentasauropus, Therapsipus tracks have been attributed to dicynodont therapsids and are restricted to Triassic strata.

The SGDS tracks described herein differ markedly from Therapsipus for much the same reasons as they differ from Pentasauropus: their digit imprint morphologies, absence of palm/sole imprints, and much smaller size all prevent referral to Therapsipus.

DicynodontipusRühle von Lilienstern, 1944 (Figs. 10F and 10G)

Dicynodontipus has been reported from “Middle” Permian–Upper Triassic strata of Brazil (Francischini et al., 2018), Italy (Conti et al., 1977), South Africa (De Klerk, 2002), Argentina (Citton et al., 2021; Marsicano et al., 2004; Melchor & De Valais, 2006), Australia (Retallack, 1996), and Germany (Rühle von Lilienstern, 1944; De Valais et al., 2020). Dicynodontipus trackways are unknown after the Triassic, although some of the questionable, Early Jurassic ichnotaxa from Lesotho named by Ellenberger have some similarities (Melchor & De Valais, 2006). Despite their name and original attribution to dicynodont therapsids (e.g., Conti et al., 1977; Retallack, 1996; Rühle von Lilienstern, 1944), Dicynodontipus tracks likely were made by cynodont therapsids (e.g., Da Silva et al., 2008; Haubold, 1971; Haubold, 1984; Marsicano et al., 2004). If correct, and if all tracks referred to this ichnogenus truly belong in it, then the temporal extent of this ichnogenus suggests that cynodont manus and pedes were evolutionarily rather conservative from the Permian through the Triassic.

Both manus and pes tracks are pentadactyl, plantigrade, mesaxonic to slightly ectaxonic, wider than long, and have short, subequal digit imprint lengths (Da Silva et al., 2008); some referred specimens are tetradactyl or tridactyl (Da Silva et al., 2008). Digit imprints are all oriented anteriorly (Melchor & De Valais, 2006). The tracks resemble those of Therapsipus but have longer, more tapering digit imprints and longer, more posteriorly extensive, convex, and rounded sole imprints (Da Silva et al., 2008; Rühle von Lilienstern, 1944). Dicynodontipus trackways also have higher pace angulations than do those of Pentasauropus or Therapsipus (Melchor & De Valais, 2006).

The SGDS tracks described herein are not pentadactyl or plantigrade, unlike Dicynodontipus tracks. Some tracks referred to Dicynodontipus have long, tapering digit imprints (e.g., Marsicano et al., 2004, fig 5); others (originally placed in the ichnotaxon Gallegosichnus by Casamiquela, 1964) have shorter, distally rounded digit imprints (e.g., Melchor & De Valais, 2006, fig 5a; Fig. 10F); and still others (originally placed in the ichnotaxon Calibarichnus by Casamiquela (1964)) have shorter, distally tapering digit imprints (e.g., Melchor & De Valais, 2006, fig 5b; Fig. 10G). Both of the latter morphologies more closely resemble those of many of the SGDS tracks described herein. Additionally, the digit imprints of the SGDS tracks are similar to those of Dicynodontipus in divarication angles and the relative degrees of anterior projection (Marsicano et al., 2004; Melchor & De Valais, 2006). However, the consistent lack of pentadactyly, mesaxony, and sole imprints prevents us from readily referring the SGDS tracks to Dicynodontipus.

Cynodontipus Ellenberger, 1976

Ellenberger (1976) described a single, incomplete fossil—ostensibly a track that includes hair imprints—from the Middle Triassic of France as Cynodontipus and attributed it to a (presumably non-mammaliaform) cynodont therapsid. Additional specimens were reported from the Middle and Late Triassic of Morocco and eastern North America (Olsen, Et-Touhami & Whiteside, 2012). Subsequent work, however, demonstrated that the type specimen is not a singular track at all, though interpretations of it vary. Olsen, Et-Touhami & Whiteside (2012) and Sues & Olsen (2015) interpreted specimens as procolophonid burrows; Klein & Lucas (2021) regarded the French specimen as a partial chirotheriid track with microbially induced sedimentary structures and the ichnotaxon as a nomen dubium. In either case, it is incomparable to any of the SGDS specimens, but it is mentioned here because it otherwise is one of only a few Mesozoic ichnotaxa to have been (albeit incorrectly) attributed to a non-mammaliaform cynodont.

BrasilichniumLeonardi, 1981 (Fig. 10H)

Brasilichnium and Brasilichnium-like tracks have been reported from primarily eolian deposits of Early Triassic–Late Cretaceous age almost globally (see Leonardi & De Souza Carvalho, 2020 for a review of occurrences). The variety of morphologies that have been attributed to this ichnotaxon suggest that, like Batrachopus, it may have become something of an ichnotaxonomic “wastebasket” (Leonardi & De Souza Carvalho, 2020). Brasilichnium and Brasilichnium-like tracks generally have been attributed to a derived synapsid (derived, non-mammaliaform eucynodont or basal mammaliaform—see discussion in D’Orazi Porchetti, Bertini & Langer (2016)); within that interpretation, the general brevity of the digit imprints in such tracks suggests that their track makers had digital arcades (sensu Kümmell & Frey, 2012), supporting a therapsid attribution. However, because most Brasilichnium and Brasilichnium-like tracks were registered on dune faces in eolian sediments, many, if not most, have been affected by extramorphological phenomena, such as sediment collapse and deformation features, as well as asymmetrical push-up rims (e.g., Engelmann & Chure, 2017; Leonardi, De Souza Carvalho & Fernandes, 2007; q.v. Loope, 2006), that create rather variable track morphologies and complicate interpretations of the manus and pes morphologies of the track makers.

The type ichnospecies, B. elusivum, as revised by Fernandes & De Souza Carvalho (2008; q.v., Buck et al., 2016; D’Orazi Porchetti, Bertini & Langer, 2018), comprises quadrupedal trackways that lack tail traces. Detailed pes prints and less common manus prints typically are wider mediolaterally than long anteroposteriorly. Both the smaller manus prints, when preserved, and the larger pes prints are ectaxonic and tetradactyl (digits II–V; digit I is always absent), but many referred specimens exhibit fewer digit imprints, or even no discreet digit imprints. When preserved on both the manus and pes, digit imprints are short and teardrop-shaped (typically rounded proximally and tapered distally), but imprints can also be rounded distally. Either digit imprints III and IV (for B. elusivum; Buck et al., 2016) or II and III (for B. anaiti; D’Orazi Porchetti, Bertini & Langer, 2018 are longest. Sole marks are rounded and usually wider than long. Manus prints tend to be located anterior to the pes prints.

The general brevity of the SGDS track digit imprints, the divarication angles, and the nearly co-equal anterior projections of the digit imprints of the tetradactyl SGDS tracks (190, 18-T3-3, 18-T3-4, 18-T7) all resemble those of Brasilichnium manus and pes tracks. Even the possible SGDS manus track (18-T3-5) proportionally bears some resemblance to a Brasilichnium manus track that lacks a digit II imprint, but its position largely lateral to its associated pes print, rather than anterior to it, is unlike Brasilichnium. Although it is not a diagnostic feature of the ichnogenus, some Brasilichnium tracks exhibit apparent paraxony, with imprints of digits II and III angled inward and digits IV and V angled outward (e.g., D’Orazi Porchetti, Bertini & Langer, 2016, fig 2), as in SGDS 18-T3-3, 18-T3-4, and 18-T7. Further comparisons between the SGDS specimens and Brasilichnium sensu stricto are limited, however, because the SGDS specimens lack sole marks, and have variably tapering or rounded distal digit imprints. Furthermore, many Brasilichnium tracks in trackways are rotated slightly inward (e.g., D’Orazi Porchetti, Bertini & Langer, 2018, fig 2), but tracks in the SGDS 18-T3 trackway appear rotated slightly outward, as are some trackways referred to Brasilichnium from elsewhere in the American Southwest (Lockley, 2011; Rowland & Mercadante, 2014). These distinctions could arise from anything from track-making-species idiosyncrasies to differing locomotory strategies, sediments, and paleoenvironments between those of typical Brasilichnium tracks (loose, coarser sands on dune slopes) and those of the SGDS tracks (fine-grained, likely wet sand on a flat lake shore). Nevertheless, of the ichnotaxa compared in this section, the SGDS tracks most closely resemble Brasilichnium.

Other possible track makers

A lepidosaurian track maker for the SGDS tracks described herein must be considered, although few ichnotaxa from the early Mesozoic have been attributed to such track makers (Rhynchosauroides tracks have been referred to as “lacertoid” and have rare Jurassic occurrences; e.g., Avanzini, Piñuela & García-Ramos, 2010; Olsen, 1988). Nevertheless, rhynchocephalians are well known from the Late Triassic and Early Jurassic (e.g., Wu, 1994) and in the stratigraphic and geographic vicinity of the SGDS (Britt et al., 2018; Simões, Kinney-Broderick & Pierce, 2022); contemporaneous lizards, although rare, are also known, though not from North America (Datta & Ray, 2006; Evans, Prasad & Manhas, 2002).

Modern rhynchocephalian tracks have not been studied, but the manus and pes of the extant Sphenodon (per Osawa, 1898), as well as those of many Mesozoic rhynchocephalians (e.g., Bever & Norell, 2017), are broadly similar to those of many lizards in proportions and might be expected to leave tracks similar to those of “conventional” lizards (see below). The poorly understood ichnotaxon Exocampe (Hitchcock, 1858) sometimes has been attributed to a rhynchocephalian (sphenodontian) track maker (e.g., Milner et al., 2011) although it may be synonymous with Batrachopus (Olsen & Padian, 1986). As described and illustrated by Hitchcock (1858), Exocampe tracks consist of long, thin, distally tapering and outwardly curving digit imprints in tetradactyl manus and pentadactyl pes prints (Fig. 10I). Manus prints appear to have more widely divaricating digit imprints than pes prints. The lengths and inconsistent curvatures of the SGDS tracks differentiate them from Exocampe, but the ichnotaxon needs redescription and further investigation.

Lizard tracks have received some systematic study (Diedrich, 2005; Fichter, 1982; Kubo, 2010; Leonardi, 1975; Padian & Olsen, 1984). Though usually pentadactyl, the diversity of extant lizard manus and pes morphologies, plus differing substrate conditions, leads to a range of track morphologies that are not always pentadactyl (Kubo, 2010). In well-preserved, “conventional” lizard tracks (Fig. 10J), the claws make the deepest imprints, followed by the long, narrow digits (Kubo, 2010). SGDS 190, lacking any indication of claw imprints, despite deep digit imprints, thus is not likely lacertilian (or rhynchocephalian—see above). Lizard manus tracks often, but not always, exhibit widely splayed divarication angles; “conventional” pes prints tend to have more parallel digit imprints except for a widely divergent digit V. “Conventional” lizard pes tracks also have marked progressions in length from digit I–IV (e.g., Kubo, 2010, figs. 4H and 4I), unlike the SGDS tracks described herein, but other lizards have more subequal digits III–IV or even II–IV (e.g., Kubo, 2010, figs. 4C–4E), more similar to the SGDS tracks described herein. Manus and pes elements are not known for any of the earliest (Late Triassic–Early Jurassic) lizards, but parsimoniously, those taxa had unspecialized, morphologically “primitive” manus and pedes and would have made “conventional” lizard tracks (sensu Kubo, 2010). Furthermore, the high pace angulation of trackway SGDS 18-T3 argues against a sprawling, lacertilian or rhynchocephalian track maker. The SGDS tracks described herein do not readily fit a “conventional” lizard or rhynchocephalian model based on the digit imprint lengths, but a lizard or rhynchocephalian with “unconventional” manus and pes morphologies cannot be absolutely ruled out, although the apparent absence of lizards in the Late Triassic–Early Jurassic of North America argues against a strictly lacertilian interpretation. Some lizard tracks can include claw drag impressions (Padian & Olsen, 1984), but these differ from the drags seen in SGDS 18-T3 by not leading to secondary imprints. Lastly, lizard tracks are sometimes, but not always, accompanied by shallow tail impressions, which are absent with any of the SGDS tracks described herein.

Drepanosaurids, a bizarre group of small, superficially lizard-like animals, are known from the Late Triassic–Early Jurassic of North America (Berman & Reisz, 1992; Britt et al., 2018; Colbert & Olsen, 2001; Harris & Downs, 2002). Some seem to have highly specialized manus and pedes (Pinna, 1984; Renesto, 1994a; Renesto, 1994b) and would not be expected to leave “conventional” lizard-like tracks. The type specimen of the American drepanosaurid Dolabrosaurus preserves a partial manus that appears to retain “conventional” lizard-like proportions and morphology (Berman & Reisz, 1992), but the manus and pes of Hypuronector are too incomplete to reveal much information (Colbert & Olsen, 2001). The Italian drepanosaurid Drepanosaurus has an enlarged ungual on digit II of the manus (Renesto, 1994a), and Megalancosaurus has a chameleon-like manus, with two digits opposed to the other three. Drepanosaurid tracks have never been reported, but none of the SGDS tracks described herein could have been made by manus with either Drepanosaurus or Megalancosaurus-like morphologies. Digits I–V of the pes of Drepanosaurus, as reconstructed by Pinna (1984), are subequal in length; the pes of Megalancosaurus is more “conventionally” lizard-like (Renesto, 1994b). That some of the isolated SGDS tracks described herein were made by a Drepanosaurus-like pes cannot be entirely ruled out, but the tracks in trackway SGDS 18-T3 are unlike the pes of Drepanosaurus in morphology. The possibility remains open that the SGDS tracks could have been made by a taxon similar to Dolabrosaurus or Hypuronector, but if these taxa had “conventional” lizard-like manus or pedes, then the same reasoning outlined above for lizards would also apply.

Chelonian body fossils are known from the Late Triassic (e.g., Gaffney, 1990; Lichtig & Lucas, 2021) and Early Jurassic (e.g., Anquetin et al., 2008; Gaffney & Jenkins Jr, 2010). Mesozoic turtle tracks are not uncommon, and are even known from the Middle and Late Triassic (De la Fuente, Sterli & Krapovicas, 2021; Lichtig et al., 2017). When not exhibiting swim-track morphology (see below), fossil tracks of terrestrial turtles generally are far wider mediolaterally than long anteroposteriorly and consist of short, tapering, subparallel digit imprints of roughly equal lengths that often connect to one another proximally (Avanzini et al., 2005; Lichtig et al., 2017; Fig. 10K). Fossil turtle tracks often consist of claw-only imprints, but palm and sole imprints have been documented for extant turtles (Tkaczyk, 2015): palm imprints are ovate and wider than long; sole imprints are also ovate, but longer than wide. Unlike in most fossil turtle tracks, claw imprints of extant turtle tracks often are separated from palm/sole imprints by short gaps (Tkaczyk, 2015), making them morphologically reminiscent of Pentasauropus tracks. Turtle tracks also have relatively low pace angulation values, reflective of their sprawling posture. Turtle tracks are unlike the SGDS tracks described herein.

The tracks described herein also could have been made by a non-amniote tetrapod (i.e., an “amphibian”). By the Early Jurassic, the temnospondyls that were abundant in the Late Triassic were virtually extinct, with rare post-Triassic representatives known only from Asia (e.g., Maisch & Matzke, 2005) and Australia (e.g., Warren & Hutchinson, 1983; Warren, Rich & Vickers-Rich, 1997); none are known from post-Triassic North America from either footprints or body fossils despite their abundance there prior to the Jurassic. This renders the likelihood of the SGDS tracks having been made by a temnospondyl unlikely. Among “amphibians”, a batrachian track maker would be more likely because urodelans are known from the Middle–Late Triassic (Schoch, Werneburg & Voigt, 2021) and anurans from the Late Triassic (Stocker et al., 2019).

Salamander tracks, though rare in the Mesozoic fossil record, have been well-studied for extant taxa across a variety of conditions (Brand, 1996; Fichter, 1982; Peabody, 1959). Pentadactyl, clawless salamanders nominally leave pentadactyl tracks with elongate digit imprints in a rather radial pattern (Fig. 10L), but under some substrate conditions (particularly sloped subaqueous mud and sand), they make tetradactyl and even tridactyl tracks with parallel digit imprints that can have tapering digit tips (Fig. 10M). In some ways, such tracks resemble both the SGDS tracks and Brasilichnium (e.g., Brand, 1996, figs. 3 and 5). However, under most substrate conditions, salamander tracks are accompanied by pronounced tail-drag impressions, which the SGDS tracks lack. Furthermore, the SGDS tracks preserved in a trackway (SGDS 18-T3) have much greater strides than actual (or even expected) salamander trackways. A urodelan track maker therefore is unlikely.

Anuran tracks are likewise rare from the Mesozoic, but they also have been poorly studied for extant taxa. Extant anuran manus prints have strong inward rotation and possess widely divaricating digit imprints of varying lengths, often with the inner- and outermost imprints at virtually 180° to one another (Tkaczyk, 2015; Fig. 10N). Anuran pes prints have more parallel digit imprints, but often of vastly differing lengths; additionally, they are often webbed (Tkaczyk, 2015; Fig. 10N). Whether Late Triassic–Early Jurassic anurans conformed to these manual and pedal morphologies is unknown because currently known specimens do not preserve the requisite material. Anuran tracks often document saltating (hopping) behavior (e.g., Lockley & Milner, 2014) that is not evident in any of the SGDS tracks, and, as with salamanders, the stride lengths of SGDS 18-T3 are greater than would be expected for a walking anuran. An anuran track maker is not indicated.

Behavior and other considerations

Because the SGDS tracks described herein consist only of short digit imprints or, in the case of SGDS 18-T3, digit-tip imprints, the possibility that they were made by track makers that were not simply walking must be considered. As above, none of the tracks described herein appear to have been made by a saltating track maker. Tracks made by running animals tend to be morphologically similar to those of walking animals; running gaits are identified more by greater stride lengths than by track morphology, so the digit- and digit-tip-only morphologies of the SGDS tracks described herein are not readily explained as a consequence of running. If the SGDS 18-T3 tracks trailed into only drag impressions anteriorly, then running would be a distinct possible interpretation. However, SGDS 18-T3 drag impressions connect deeper tracks to virtually identical, shallower tracks only a short distance away; we find it unlikely that a running animal would create such double-tracks in close proximity.

The possibility that the SGDS tracks are swim tracks made by a water-buoyed track maker must also be considered. The SGDS collection includes the largest collection of theropod dinosaur swim tracks (Characichnos) currently known (Milner, Lockley & Kirkland, 2006), although these are not from the Top Surface of the Johnson Farm Sandstone Bed. Swim tracks of other sauropsids are also known from the Washington County, Utah area as well (Lockley & Milner, 2006). Typical swim tracks, across terrestrial, limbed tetrapod track makers, have highly distinctive morphologies consisting of sets of elongate, straight to wavy, tapering, parallel impressions (e.g., Diaz-Martínez, Citton & Castanera, 2023; Diedrich, 2005; Ezquerra et al., 2007; Lockley et al., 2021a; Lockley et al., 2021b; Milner & Lockley, 2016; Milner, Lockley & Kirkland, 2006; Thomson & Droser, 2015; Vila et al., 2015), unlike any of the SGDS tracks described herein. A few Batrachopus trackways on the Top Surface at the SGDS include transitions from swim to walk and walk to swim as they enter or exit water-filled swales in the Top Surface (Milner, Lockley & Kirkland, 2006), but the swim tracks in these examples are of the expected morphology; all other tracks on the Top Surface were made by walking animals, plus two running theropods. Close to the in situ tracks described herein are normal Batrachopus tracks and a normal Grallator track comparable in size to the possible eucynodont tracks. The tracks described herein do not exhibit swim track morphology and were not made by swimming animals.

The SGDS tracks described herein could also be undertracks, which could explain their incompleteness (e.g., the absence of sole imprints). When the SGDS 18 surface was being excavated, however, no tracks on immediately overlying layers were noted. While they could have been missed, they were not likely more than a few millimeters above the SGDS 18 surface, so if any of them are undertracks, they likely still preserve most of the morphology of the track-making appendages. Alternatively, the small size of the tracks implies a small, lightweight track maker that simply may not have registered deep prints at all, especially if the substrate was relatively firm at the time the tracks were registered. We consider this option more likely than that the tracks are deep undertracks.

Discussion

The enigmatic SGDS tracks described herein cannot be conclusively assigned, or even referred, to any established ichnotaxon or ichnotaxa. However, they are too few in number, and too incompletely preserved, to warrant establishing a new ichnotaxon for them. Based on digit-imprint morphology and overall preserved track morphology, they are better attributed to a derived synapsid (eucynodont or mammaliaform) track maker than to any of the crocodyliforms or dinosaurs well established as makers of other tracks common at the SGDS, in the region, and in the earliest Jurassic. Similarly, the tracks do not align with either the manus and pes morphologies or known tracks of several other vertebrates that were almost certainly present in this region in the earliest Jurassic (e.g., rhynchocephalians, turtles). Morphologically overall, the SGDS tracks described herein share more in common with Brasilichnium than with any other ichnotaxon; they also resemble some tracks referred to Dicynodontipus, but less so than Brasilichnium for reasons detailed above. Both of these ichnotaxa have been attributed to cynodont track makers. Brasilichnium and Brasilichnium- like tracks, however, are known primarily from coarser, eolian sandstone deposits, whereas the SGDS tracks were made in wet, lake-shore, fine-grained sand, as were many tracks referred to Dicynodontipus (Da Silva et al., 2008; Melchor & De Valais, 2006). If the SGDS tracks were made by similar track makers as were Brasilichnium and Dicynodontipus, then the morphological differences between them may reflect substrate, environmental, and/or behavioral differences. Nevertheless, the SGDS tracks constitute rare instances of Early Mesozoic—specifically, post-Triassic—synapsid tracks outside of an eolian paleoenvironment (i.e., in the Eubrontes ichnocoenose of the Grallator ichnofacies, rather than in the Brasilichnium ichnocoenose of the Chelichnus ichnofacies Hunt & Lucas, 2006a; Hunt & Lucas, 2006b). They raise the question of what Brasilichnium track-maker tracks made outside of eolian settings might look like—specifically, if they might look like Dicynodontipus tracks. If both Brasilichnium and Dicynodontipus were made by cynodonts, further comparisons of tracks referred to each ichnotaxon would test whether they were either (1) made by track makers with similar manual and pedal morphologies, and their differing track morphologies stem largely from environmental factors (meaning that the two ichnogenera could be synonyms), or (2) made by taxa with distinct manual and pedal morphologies (supporting ichnogeneric distinction). This question is, however, beyond the scope of this paper.

Non-mammaliaform eucynodont or mammaliaform track makers for the SGDS tracks described herein are plausible for two reasons:

1. Tracks (specifically Brasilichnium) attributed to such animals have been reported from Lower Jurassic strata elsewhere in the American Southwest (Engelmann & Chure, 2017; Hamblin & Foster, 2000; Lockley, 2011; Lockley et al., 1998; Lockley et al., 2014; Reynolds, 2006; Rowland & Mercadante, 2014); and

2. Derived, non-mammaliaform eucynodont (specifically tritylodontid) and basal mammaliaform skeletal materials are known from both Upper Triassic and Lower Jurassic strata in the American Southwest, albeit not yet from the Moenave Formation, or from Utah. Derived, non-mammaliaform eucynodonts (Kligman et al., 2020) and basal mammaliaforms (Lucas & Luo, 1993) are known from the Upper Triassic Chinle Formation of northern Arizona and the Tecovas Formation (Dockum Group) of west Texas. Similarly, and more abundantly, derived, non-mammaliaform eucynodonts (Hoffman & Rowe, 2018; Kermack, 1982; Lewis, 1986; Sues, 1985; Sues, 1986; Sues, Clarke & Jenkins Jr, 1994; Sues & Jenkins Jr, 2006) and basal mammaliaforms (Crompton & Luo, 1993; Jenkins Jr, Crompton & Downs, 1983; Sues, Clarke & Jenkins Jr, 1994) are known from the Lower Jurassic Kayenta Formation of northern Arizona.

The Chinle and Kayenta formations stratigraphically bracket the Moenave Formation (Fig. 2), so the presence of similar taxa in southwestern Utah during Moenave Formation time can be assumed. However, known skeletal material of the aforementioned early Mesozoic basal mammaliaforms does not, as yet, include manual or pedal material, so how the manus and pedes of these taxa might align with the SGDS tracks cannot be determined. Skeletal material of the Kayenta Formation tritylodontid Kayentatherium includes manual (Hoffman & Rowe, 2018; Sues & Jenkins Jr, 2006) and pedal (Lewis, 1986) material; an indeterminate (per Sues, Clarke & Jenkins Jr, 1994), partial tritylodontid skeleton from the uppermost Kayenta Formation and overlying Navajo Sandstone also preserves manual material (Milner et al., 2023a; Milner et al., 2023b; Winkler et al., 1991). Relative digital proportions in these non-mammaliaform eucynodonts have not been described, however. All are pentadactyl, so understanding their digital proportions and locomotory postures relates directly to whether or not they could be predicted to have normally made pentadactyl or tetradactyl tracks and what the nominal, expected relative lengths and projections of their digit imprints would be, even incorporating digital arcades (sensu Kümmell & Frey, 2012).

Ichnological evidence of a derived, non-mammaliaform eucynodont or basal mammaliaform (or, conceivably, both) in the Whitmore Point Member of the Moenave Formation of southwestern Utah increases the known ichnofaunal, and consequent faunal, diversity of the unit and the region. At the SGDS specifically, tetrapod ichnofossils are, as present, otherwise limited to those of crocodyliforms (Batrachopus), dinosaurs (Anomoepus, Characichnos, Eubrontes, Gigandipus, Grallator, Kayentapus), and possibly rhynchocephalians (Exocampe) (Harris & Milner, 2015; Milner et al., 2011). Additionally, a diverse invertebrate ichnofauna is also present (Lucas et al., 2006; Rose, Harris & Milner, 2021), as are Undichna fish swim trails (Milner et al., 2011). The Whitmore Point Member thus ichnologically preserves a detailed “snapshot” of an earliest Jurassic terrestrial ecosystem that now likely includes either or both non-mammaliaform or mammaliaform eucynodont therapsids.

Future excavations of the Top Surface and other track-bearing horizons at the SGDS potentially may uncover more tracks that may clarify the nature of the enigmatic SGDS track maker(s) and the ichnotaxon/ichnotaxa to which the tracks described herein pertain. Additionally, further work exploring the effects of substrate differences and/or slope angles on synapsid track formation is needed. This could constitute digital modeling and/or experimentation with extant mammals that possess manual and pedal morphologies similar to those of derived, non-mammaliaform eucynodonts and basal mammaliaforms. Controlled experimentation to determine the relationship between substrate characteristics and track formation has become more common for invertebrate ichnological track work (e.g., Azain, 2006; Davis, Minter & Braddy, 2007; Fairchild & Hasiotis, 2011; Schmerge, Riese & Hasiotis, 2013), but for vertebrates, similar work, while invaluable, often has been observational in natural, rather than controlled, conditions (e.g., Farlow & Elsey, 2010; Farlow et al., 2017; Genise et al., 2009; but see Buck et al., 2016; Leonardi, 1982, Marchetti et al., 2019, Milàn, 2006, and Turner & Gatesy, 2021 for experimental examples). This undoubtedly is due to difficulties in handling and controlling the locomotory behaviors of (especially larger) vertebrates. Nevertheless, small mammals might be manageable for such experimental work (e.g., Buck et al., 2016).

Conclusions

Several enigmatic, partial fossil tracks from the Lower Jurassic Whitmore Point Member of the Moenave Formation at the SGDS are most likely attributable to a eucynodont (derived, non-mammaliaform or basal mammaliaform) track maker. The tracks described herein include a single possible manus track along with several pes tracks. The possible manus is tridactyl with only the digits imprints preserved that taper distally. The pes prints are tetradactyl with only digits imprints preserved. Some taper proximally and/or distally, but others are rounded on both ends. These tracks are most similar among known and contemporaneous ichnotaxa to Brasilichnium, which is widely understood to pertain to a derived, non-mammaliaform or basal mammaliaform track maker. However, the incomplete nature of the SGDS tracks prevents us from assigning the tracks to this ichnotaxon. Morphological differences between the SGDS tracks and those of Brasilichnium may be functions of substrate differences: the former were made in probably wet, fine-grained, lake-shore sands, whereas the latter are known only from coarser, eolian sands. The discovery of further, more complete specimens and/or experimental work to better establish a relationship between substrate and track morphology are needed to clarify the nature of the SGDS tracks.

Supplemental Information

Supplemental Information 1 Three-dimensional model, produced by laser scan, of possible eucynodont track SGDS 190

For details on the production of this figure, please see the Materials and Methods.

Spencer Lucas (New Mexico Museum of Natural History & Sciences) was the first to suggest the possibility of these tracks pertaining to cynodonts. Jaleesa Buchwitz performed the photogrammetry, and Rob Gay performed the laser scan of SGDS 190 that appears in the supplementary material. We thank Heitor Francischini (Universidade Federal do Rio Grande do Sul) and Verónica Krapovickas (Friedrich-Alexander-Universität Erlangen-Nürnberg) for the taking the time to make valuable comments that greatly improved the paper. We also thank the SGDS for allowing us to use their resources.

Additional Information and Declarations

Competing Interests

Author Contributions

Data Availability

The authors declare there are no competing interests.

Holly Hurtado performed the experiments, analyzed the data, prepared figures and/or tables, authored or reviewed drafts of the article, and approved the final draft.

Jerald D. Harris performed the experiments, analyzed the data, prepared figures and/or tables, authored or reviewed drafts of the article, and approved the final draft.

Andrew R.C. Milner conceived and designed the experiments, analyzed the data, authored or reviewed drafts of the article, and approved the final draft.

The following information was supplied regarding data availability:

The tracks described herein all come from the Top Surface horizons and are in situ, except SGDS 190, which is ex situ.

A replica of SGDS 190 is reposited in the University of Colorado Museum in Boulder, Colorado as specimen UCM 177.37; all other specimen replicas are retained at the St. George Dinosaur Discovery Site (SGDS).

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
