# Peer review of "Possible eucynodont (Synapsida: Cynodontia) tracks from a lacustrine facies in the Lower Jurassic Moenave Formation of southwestern Utah"

_PeerJ, doi:10.7717/peerj.17591_

## Round 0.1 · original submission · Major Revisions

Dear Holly Hurtado

I am writing you in relation to your Ms (#87860) entitled “Possible eucynodont (Synapsida: Cynodontia) tracks from a lacustrine facies in the Lower Jurassic Moenave Formation of southwestern Utah” co-authored with Jerald D. Harris and Andrew R.C. Milner.

The manuscript has been reviewed by two reviewers and myself and after a careful evaluation I consider your manuscript needs Major Revision before being considered for publication in our journal.

The main concern shared with both reviewers is the ichnotaxonomic assignation of the tracks and therefore the trackmaker identification, which is the main goal of your contribution. In fact, they are giving you an alternative hypothesis that would be important for your manuscript to explore in detail in the Discussion section. In this context, the reviewers are suggesting you improve several sections of the manuscript.

As pointed out by Reviewer #1 the environmental characterization of the track-bearing beds (Geological Setting) would be important to be discussed in detail, particularly related to the possibility that the tracks were produced underwater. Reviewer #2 also suggests you add a more in-depth anatomical comparison and to explore the locomotor behavior of the trackmaker in order to better support the trackmaker identification. Also, as mentioned by Reviewer #2, please explore the possibility that the footprints were produced by a trackmaker unknown in the body fossil record of the unit in order to maintain objectivity in your interpretations.

Finally, the figures need some improvement. The map in Figure 1 needs coordinates and scale, and a more detailed geological map of the area where the section in Figure 2 was measured. As also mentioned by Reviewer #1, in several figures one of the pair in the stereophotographs are out of focus, in particular the left one in Figure 4, the A in Figure 6, and also the left pics in Figures 9 and 10.

I am requesting you to revise your manuscript according to the detailed reviews provided and also included in an annotated version of the manuscript by Reviewer #2. Please, take particular attention to the points mentioned above.

As the revisions required are extensive enough, another round of reviews may be necessary when you resubmit your revised version.

Thank you for submitting your manuscript to PeerJ and I look forward to receiving your revision.

Sincerely,

·

Basic reporting

No comment

Experimental design

No comment

Validity of the findings

No comment

Additional comments

- The Geological Setting section can be improved with a proper description and characterization of the lacustrine bed in which the tracks occur. It will be used for strengthening some of the arguments that the authors can use for ichnotaxonomical purposes and attribution to a biological taxon (see below);
- Material and Methods can also be improved, with a mention of the parameters used for ichnotaxonomy and attribution to a biological taxon. Also, mention if the information from ichnotaxa used for comparisons were gathered from bibliography or analyzed by first hand. The whole first paragraph of the Description section can be moved from Material and Methods. Lastly, photogrammetry is not mandatory, but highly recommended;
- The description of the tracks is good and easily understandable. However, I disagree of most of the interpretation of the ichnotaxonomical assignement and attribution to a producer. As well described by the authors, all the tracks are tri- to tetradactyl (never pentadactyl) and ectaxonic; they lack digital pads, sole/palm impressions and claw marks; the digits are often curved (Fig. 9) and associated to drag marks (Figs. 5 and 7) lack of soles/palms. All these mentioned features are commonly seen in amphibian (=temnospondyl) tracks, such as Batrachichnus and, especially, Limnopus, so it would be worth to better compare these ichnotaxa to the tracks from the St. George Dinosaur Discovery Site.
- Finally, triy to tetradactyly is a common feature among tracks produced underwater (e.g., Hatcherichnus, Batrachopus etc.) because in this scenario the body of the producer is sustained by the water column (water thrust) and only part of the digits touches the ground and register the track. Therefore, it sounds reasonably to found amphibian tracks in such lacustrine environment (so, this is why is important to stress the description of the track-bearing bed) and age (Early Jurassic). I think this possibility should be better explored in the manuscript.

·

Basic reporting

Firstly, I found your study on the new footprints from the St. George Dinosaur Discovery Site (SGDS) in southwestern Utah quite intriguing, particularly in its implications regarding the trackmaker. However, I believe there are certain aspects that warrant further exploration and discussion:

Goal of the Research: The introduction of the manuscript should explicitly state the research goals and questions to provide context for evaluating the results.

Concerning other basic reporting issued, the language is clear, unambiguous, and professional. The article is well structured, as well as figures and tables.

The references are sufficient however I have suggested a few additions that may help improving the manuscript.

Experimental design

1. Trackmaker Identification: Have you considered the possibility that the morphology of the trackway T18-T3 could be attributed to a crocodyliform from the same locality (Batrachopus) using an infrequent gait, rather than a new, Eucynodont trackmaker? Given the long stride length, it suggests a fast gait. To enhance the depth of your analysis, you might want to consider estimating the potential speed of the trackmaker. Since the trackmaker remains unknown, you could calculate the speed within the context of multiple potential trackmakers.
2. Digitigrade Characteristics: You correctly pointed out that the tracks appear to be produced by a digitigrade animal. However, this digitigrade impression might be a result of the animal employing a fast gait. The presence of digit drag marks associated with the digit impressions further suggests a fast gait. It would be beneficial to delve deeper into the implications of this digitigrade nature and its potential relation to speed.  
3. Comparative Analysis: It might be worthwhile to reference recent work by Kim et al. (2020) in Scientific Reports (https://doi.org/10.1038/s41598-020-66008-7), which presented evidence of bipedal crocodylomorph trackways assigned to Batrachopus grandis ichnosp. nov. The similarities between the Batrachopus footprints of Korea and the material you presented here are striking and could provide valuable context for your findings.
4. Bipedal Arrangement: Finally, you mentioned the apparent bipedal arrangement of trackway T18-T3, which doesn't align with the hypothesis of an Eucynodont trackmaker. It would be beneficial to discuss this incongruity in more detail and consider alternative interpretations or hypotheses.  
5. Further anatomical comparison with body fossil autopodium. Another issue that needs to be addressed is the need for a more comprehensive anatomical comparison. While your study's findings are intriguing, they lack a robust anatomical context that would significantly enhance their significance and relevance. Incorporating a more detailed anatomical analysis would not only strengthen your argument but also provide a more solid foundation for your conclusions.

Validity of the findings

Instances of circular thinking in your manuscript. It is essential to ensure that your reasoning and logic are clear and do not lead to circular arguments. Please review your manuscript carefully to identify and rectify any circular reasoning to improve the overall clarity of your work. In the discussion, please review the reasoning leading to what seems to be the main results of the manuscript: the assignment of the trackmaker. It seems mostly to involve the previous interpretation of the producer of Brasilichnium, the resemblance of the studied material to it, and the record of non-mammalian Eucynodonts in correlative geographically and  stratigraphic units. These reasoning prevent the possibility of the footprints being made by other animals, still not recorded locally by the bone record, and tie the ichnological record to the bone record.

Additional comments

Firstly, I found your study on the new footprints from the St. George Dinosaur Discovery Site (SGDS) in southwestern Utah quite intriguing, particularly in its implications regarding the trackmaker. However, I believe there are certain aspects (mentioned above) that warrant further exploration and discussion.

I believe addressing these points will enhance the depth and comprehensiveness of your manuscript, making it an even more valuable contribution to the field. I look forward to seeing the revised version of your manuscript and hope that my suggestions prove helpful in refining your research.

If you have any questions or require further clarification regarding these comments, please do not hesitate to reach out.

Best regards,

Verónica Krapovickas
[email protected]

---

## Round 0.2 · Minor Revisions

Dear Holly Hurtado

I am writing you in relation to your Ms (#87860) entitled “Possible eucynodont (Synapsida: Cynodontia) tracks from a lacustrine facies in the Lower Jurassic Moenave Formation of southwestern Utah” co-authored with Jerald D. Harris and Andrew R.C. Milner.

The manuscript has been re-reviewed by Verónica Krapovickas and myself and after a careful evaluation I consider your manuscript needs Minor Revision before being considered for publication in our journal.

We agree that the new revision greatly improved the previous version, particularly the Comparisons section, and we also agree to relate the material to a non-mammalian therapsid trackmaker.

We also concur that the lack of enough characters in your material enables it to be clearly attributed to a specific ichnotaxon. Nevertheless, your material is consistently considered more similar to Brasilichnium along the Ms (Abstract, Discussion, Conclusions) to any other ichnotaxon. Our main concern is related to this consideration as you did not make any strong case against to a Dicynodontipus affinity. Also, as the reviewer pointed out, you should take into account in your discussion the implications of the contrasting depositional were your material was preserved and the eolian facies where Brasilichnium is consistently preserved.

I suggest that you consider to include also Dicynodontipus affinities together with Brasilichnium as it is more consistent, at present, with the information available in your material.

Finally, the map in Figure 1 still needs to be improved. You need to add a more detailed map of the locality as the maps you are providing has a so large scale that are not informative. You need to provide a more detailed map about the locality and the section in Figure 2 was measured.

Thank you for submitting your manuscript to PeerJ and I look forward to receiving your revision.

Sincerely,

Claudia Marsicano

·

Basic reporting

The writing is clear and professional, and references and background information are complete and updated.

Experimental design

The description of the material is extensive and complete and the comparison with other known fossil footprints is also extensive.

the methods are well explained.

Validity of the findings

The work presents novel findings.

I find the comparison between the SGDS material and Brasilichnium to be somewhat flawed. Firstly, the footprints originate from contrasting sedimentary environments—eolian dunes for Brasilichnium versus marginal lacustrine for SGDS—leading to distinct substrate deformation patterns. Brasilichnium is typically characterized more by trackway patterns and substrate deformation rather than detailed digital impressions, which contrasts with the distal digital impression, with almost no deformation of the substrate and digit drag marks often observed in SGDS material. In the rare instances where digital patterns are discernible in Brasilichnium, they usually present as 4-digit impressions, with the central digits being of similar lengths, unlike the longer digit IV impressions seen in SGDS material.

However, I do concur with the proposed interpretation of a non-mammaliaform eucynodont or mammaliaform track maker for the SGDS tracks. This agreement isn't due to any perceived similarity to Brasilichnium but rather stems from the specific features present in the SGDS tracks themselves.
I recommend reinforcing or emphasizing the intrinsic traits exhibited by the trackways

Additional comments

Line 43-44. To clarify the statement “Ichnologically, Early Mesozoic synapsids thus appear to have preferentially inhabited eolian environments.” is very important because it provides the idea of preferred habitat by Early Mesozoic synapsids, which I believe is not what the researchers meant to say and also it is not properly proven.

What I have poorly stated (sorry about that) in the previous review is that:

“Eolian environments seem to record preferentially footprints of early Mesozoic synapsids” and not the other way around, “Early Mesozoic synapsids thus appear to have preferentially inhabited eolian environments.” In the last case, it is inferred the preferred habitat (deserts) of a clade (synapsids). In the first case, it is mentioned that one environmental system (eolian systems) tends better to preserve the footprints of one clade (synapsids). At first glance may look like similar statements but they are not.

Another way to say it is:

“Early Mesozoic synapsid footprints are dominant components in the trace fossil record of eolian environments.”
In this case, you a not inferring the synapsids' preferred habitat but accounting for the footprints that are most frequently found in eolian systems.

Line 43-44. On the other hand, the expression “Early Mesozoic Synapsids” is rarely found in the literature. It would be good to define the group as you did in lines 132-134 the first time is used so it is clear to the reader from the beginning of the text.

By a search on Scholar Google, I found just one result contrasting with 4700 by looking for the word “synapsids”. It seems not to reflect a well-defined taxonomic group. Other options to name the animal group and time spam you are referring to are Late Triassic- Early Jurassic synapsids / nonmammalian synapsids / early mammaliaforms

Lines 382 – 383. Argentina is mentioned twice in the sentence, both references can be included after the first

Lines 596-597. You can also mention de la Fuente et al., 2021 for a revision of the Early and Middle Triassic turtles. https://doi.org/10.1016/j.jsames.2020.102910

Lines 645-646. “…running gaits are identified more by greater stride lengths than by track morphology, so the digit- and digit-tip-only morphologies of the SGDS tracks described herein are not readily explained as a consequence of running.”
The trackway shown in Figure 7, though may be incomplete, shows long stride length and digit drag marks in T3-3, T3-4, T3-5. That is suggestive of a fast gait, maybe a canter, (see Do tracks yield reliable information on gaits? - Part 1: The case of horses 10.5194/fr-17-59-2014) I suggest evaluating that option carefully.

Lines 678- 680. “Based on digit-imprint morphology and overall preserved track morphology, they are better attributed to a synapsid track maker than…”
Please be more specific, given that synapsids also include “pelycosaurs-grade synapsids” with very different foot morphologies. You provided a more specific assignation in line 702 “Synapsid—specifically non-mammaliaform eucynodont or mammaliaform—"

Figure 10 is an excellent figure that greatly facilitates observing the characters of the described footprints and their comparison. Unfortunately, Brasilichnium, that ichnotaxon you refer to as the most similar to the SGDS tracks is not included on the graph. It would be of great value if you could include it.

---

## Round 0.3 · Minor Revisions

Dear Dr. Hurtado

During the final revision of your Ms by the Section Editor, it was apparent that your Ms lacks any reference about data availability of the studied material (e.g. casts, slabs, digital raw data). The information about where is the studied material and digital data deposited should be included and linked to the Ms.

Hoping to hear from you soon,
Best regards,
Claudia Marsicano

---

## Round 0.4 · accepted · Accept

Dear Dr Harris
Many thanks for your quick response. I consider that now your Ms is in the adequate format for publication in PeerJ.
Thanks for your submission.
best regards
Claudia